# MAP3Kε1/2 Interact with MOB1A/1B and Play Important Roles in Control of Pollen Germination through Crosstalk with JA Signaling in Arabidopsis

**DOI:** 10.3390/ijms23052683

**Published:** 2022-02-28

**Authors:** Juan Mei, Pengmin Zhou, Yuejuan Zeng, Binyang Sun, Liqun Chen, De Ye, Xueqin Zhang

**Affiliations:** 1State Key Laboratory of Plant Physiology and Biochemistry, College of Biological Sciences, China Agricultural University, Beijing 100193, China; sz20173020125@cau.edu.cn (J.M.); zhoupengmin@genetics.ac.cn (P.Z.); sz20163020123@cau.edu.cn (Y.Z.); s20203020244@cau.edu.cn (B.S.); chenliqun@cau.edu.cn (L.C.); yede@cau.edu.cn (D.Y.); 2State Key Laboratory of Molecular Developmental Biology, Institute of Genetics and Developmental Biology, Chinese Academy of Sciences, Beijing 100101, China

**Keywords:** MAP3Kεs, MOB proteins, JA, precocious germination, Arabidopsis

## Abstract

Restriction of pollen germination before the pollen grain is pollinated to stigma is essential for successful fertilization in angiosperms. However, the mechanisms underlying the process remain poorly understood. Here, we report functional characterization of the MAPKKK kinases, MAP3Kε1 and MAP3Kε2, involve in control of pollen germination in Arabidopsis. The two genes were expressed in different tissues with higher expression levels in the tricellular pollen grains. The *map3kε1 map3kε2* double mutation caused abnormal callose accumulation, increasing level of JA and precocious pollen germination, resulting in significantly reduced seed set. Furthermore, the *map3kε1 map3kε2* double mutations obviously upregulated the expression levels of genes in JA biosynthesis and signaling. The MAP3Kε1/2 interacted with MOB1A/1B which shared homology with the core components of Hippo singling pathway in yeast. The Arabidopsis *mob1a mob1b* mutant also exhibited a similar phenotype of precocious pollen germination to that in *map3kε1 map3kε2* mutants. Taken together, these results suggested that the MAP3Kεs interacted with MOB1s and played important role in restriction of the precocious pollen germination, possibly through crosstalk with JA signaling and influencing callose accumulation in Arabidopsis.

## 1. Introduction

During anthesis of flowering plants, the mature pollen grains are released from the dehiscent anther and delivered to the stigma of the pistil. After its recognition with the stigmatic cells, the pollen grain hydrates and germinates, generating a pollen tube, which further carries the two sperm cells into the embryo sac for double fertilization [1,2]. Therefore, pollen germination and pollen tube growth are essential for successful sexual reproduction in angiosperm plants. Defects in pollen grain germination or pollen tube growth will cause male sterility. Many factors have been identified as the positive regulators of pollen germination and pollen tube growth, such as calmodulin, Ca^2+^, phosphatidylinositol, H^+^, NO, protein kinase, ROS, pH, etc. [3,4,5,6,7]. Studies also show that restriction of pollen germination before the pollen grain is pollinated onto the stigma, is also significant for successful fertilization. Loss of the germination-restrictive functions will cause precocious germination of the pollen grains in the anthers before they are pollinated to the stigma, leading to failure of fertilization. For example, the *raring-to-go* (*rtg*) mutant pollen grains exhibit precocious germination in the anthers at the bicellular pollen stage due to aberrant callose deposition in the pollen grains [8]. Furthermore, Arabidopsis *plantacyanin* overexpression- results in abnormal callose deposition in pollen walls and leads to a few of the transgenic pollen grains precociously germinating in the indehiscent anther [9]. These results and more and more newly identified evidence show that the abnormal callose deposition is related to precocious germination of pollen grains. The direct evidence is achieved from the characterization of callose synthases. Both mutations in *Glucan Synthase*-*Like 10*/*Callose Synthase 9* (*GSL10*/*Cals9*) and overexpression of *Glucan Synthase*-*Like 2*/*Callose Synthase 5* (*GSL2*/*Cals5*) could cause aberrant callose deposition in bicellular pollen grains which precociously germinate inside the anthers [10]. In addition, several genes, which are involved in synthesis of the important substances and hormones are also important for controlling the precocious pollen germination. The Arabinogalactan Proteins (AGPs) are important components of the pollen walls. Loss of *AGP6*, *AGP11*, and *fasciclin*-*like AGPs* (*FLAs*) cause defects in pollen development and germination [11,12]. The *nicotinate/nicotinamide mononucleotide adenyltransferase* (*NMNAT*) is a key enzyme for NAD biosynthesis. Mutation in *NMNAT* leads to precocious pollen germination in anthers [13]. Moreover, the *GTP*-*Binding Protein Related1* (*GPR1*) is also involved in the restriction of pollen precocious germination, and a loss of its function causes precocious pollen germination at the triple cellular stage in the anthers [14].

JA is an important hormone for plant growth, which also regulates anther development and pollen maturation [15,16]. Mutations of the genes involved in the JA biosynthesis and signaling pathway can affect both vegetative and reproductive organ development in plants. For example, the *fatty acid desaturation* (*fad*) [17], *coronatine insensitive protein 1* (*coi1*) [18], *2-oxophytodienoic acid reductase* (*opr3*) [19], *delayed-dehiscence1* (*dde1*) and *dde2* [20,21], *defective in anther dehiscence 1* (*dad1*) [22], and *allene oxide synthase* (*aos*) [23] are all defective in reproductive organ development. Methyl jasmonate (MeJA) influences pollen germination and pollen tube elongation of *Pinus nigra* in vitro [24]. Arabidopsis *DAYU* (*DAU*) encodes the peroxisome membrane protein ABERRANT PEROXISOME MORPHOLOGY 9 (APEM9), which regulates pollen germination by participating in JA biosynthesis [25]. JINGUBANG (JGB) containing the WD40 domain regulates JA biosynthesis via interaction with TEOSINTE BRANCHED1/CYCLOIDEA/PROLIFERATING CELL NUCLEAR ANTIGEN FACTOR 4 (TCP4)transcription factor, affecting pollen germination, indicating that high-level JA could promote pollen germination [26]. Taken together, restriction of precocious pollen germination before the pollen grains are pollinated on the stigma is controlled by a complex mechanism, which remains largely unknown.

The Arabidopsis MAP3Kε1 and MAP3Kε2 (MAP3Kεs, MAP3Kε1/2) have been catalogued to the MAPK protein kinase family [27]. The phylogenetic analysis shows that the Arabidopsis MAP3Kεs share the closest similarity to the Cdc7 from *Schizosaccharomyces pombe* and Cdc15 from *Saccharomyces cerevisiae* [28]. Cdc7 is a component of the septation initiation network (SIN) which regulates septum formation after chromosome segregation in cell division [29]. The protein kinase Cdc15 phosphorylates and activates the downstream Dbf2/20-Mob1 complex to participate in mitotic exit network (MEN) [30]. The core components of the Hippo pathway and MEN/SIN pathway are strongly conserved from yeast to humans, including the upstream factor Germinal Center Kinases (GCK/Ste20) protein kinase, the downstream NDR protein kinase, and the NDR co-activator MOB protein [31,32]. There are few reports on the Hippo signaling pathway in plants. For example, SIK1, a Hippo/STE20 homolog can interact with MOB1A and MOB1B, and regulates cell proliferation and expansion in Arabidopsis [33]. *MOB1A* is required for sporophyte and gametophyte development and plays a critical role in auxin-mediated plant development [34,35,36]. *MOB1A* and *MOB1B* also regulate JA accumulation and root development [37]. Recently, three NDR kinases, NDR2/4/5, and MOB1A/1B, which function redundantly, are identified as being required for preventing precocious pollen germination [38]. This result indicates that the core components of Hippo pathway and MEN/SIN pathway in plants are also likely to be involved in the control of pollen precocious germination. However, their signaling mechanisms underlying the restriction of pollen germination remain unclear.

In this study, we performed functional characterization of the Arabidopsis proteins MAP3Kε1 and MAP3Kε2, which were homologous to Cdc15 in yeast. The *map3kε1/+*; *map3kε2*/−, and *map3kε1*/−; *map3kε2*/+ exhibited abnormal pollen development and precocious pollen germination in the dehiscent anthers. Part of the mature mutant pollen grains had abnormal callose deposition. Furthermore, the expression levels of many JA-related genes were upregulated in the *map3kε1*/+; *map3kε2*/− and *map3kε1*/−; *map3kε2*/+ mutants, consistent with the increase of the JA content in the mutants. The protein–protein interaction assays, including yeast two-hybrid (Y2H), Co-Immunoprecipitation (Co-IP), and luciferase complementation image (LCI) assays, showed that MAP3Kε1/2 could interact with MOB1A/1B, respectively. Moreover, genetic analysis indicated that the phenotypes of the *map3kε1*/−; *map3kε2*/+; *mob1a*/+; *mob1b*/− and *map3kε1*/+; *map3kε2*/−; *mob1a*/+; *mob1b*/− were similar to that of the *mob1a*/+; *mob1b*/− mutant. Together, our study suggested that the MAP3Kεs may interact with the MOB1s and regulated pollen germination through influencing JA homeostasis in Arabidopsis pollen.

## 2. Results

### 2.1. MAP3Kε1/2 Were Expressed Ubiquitously in Different Tissues

Phylogenetic analysis showed that the MAP3Kεs existed widely in both monocotyledons and dicotyledons, such as *Zea mays*, *Oryza sativa*, and *Glycine max*, and had characteristics similar to those of Cdc15 and Cdc7 in yeast. Further structural analysis indicated that MAP3Kε1/2 had conserved N-terminal kinase domain that shared high homology with the domain in STKc_Cdc7_like protein kinase (Appendix A). Quantitative real-time PCR (qRT-PCR) and promoter activity assays were further performed to investigate their expression. The results of qRT-PCR showed that *MAP3Kε1/2* were expressed widely in different Arabidopsis tissues, especially higher in mature pollen grains and siliques (Appendix A). For promoter::*GUS* activity assays, the two promoter fragments of *MAP3Kε1*/*2* were fused with the *GUS* reporter gene in Ti-derived vector pCAMBIA1300 and introduced into wild-type plants, respectively. Consistent with qRT-PCR results, GUS activity was detected in different tissues of the p*MAP3Kε1*::*GUS* and p*MAP3Kε2*::*GUS* transgenic plants. The higher GUS activities also were detected in mature pollen grains (Appendix A). These results suggested that *MAP3Kε1* and *MAP3Kε2* may play wide spectrum of roles in different Arabidopsis tissues, including pollen development and pollen tube growth.

### 2.2. The map3kε1 map3kε2 Double Mutant Enhanced Pollen Germination

Previous study showed that the single mutants generated by T-DNA insertion in either *MAP3Kε1* or *MAP3Kε2* displayed normal development, while the double mutant combination showed abnormal pollen development [39], indicating that *MAP3Kε1* and *MAP3Kε2* functioned redundantly and played important roles in pollen development. To further investigate the functions of *MAP3Kε1*/*2* in plant reproductive development, we obtained the mutants from the Arabidopsis Biological Resource Center (ARBC). In the SALK_133360C, T-DNA was inserted in the 17th exon of *MAP3Kε1*, 4534 bp downstream of ATG. In the SALK_084747C, T-DNA was inserted in the 6th intron of *MAP3Kε2*, 969 bp downstream of ATG (Appendix A). The sites of T-DNA insertion were further confirmed by PCR using gene-specific and T-DNA border primers. The RT-PCR results indicated that the T-DNA-insertion mutants were homozygous (Appendix A). Neither *MAP3Kε1* nor *MAP3Kε2* transcript was detected in the *map3kε1*/−; *map3kε2*/+ plants or *map3kε1*/+; *map3kε2*/− plants (Appendix A), respectively. Furthermore, both the *map3kε1* and *map3kε2* plants showed the same growth patterns as normal as that of wild-type plants. Then, the double mutants were generated by crosses of the single mutants. No double homozygous mutant plant was obtained. Therefore, two types of heterozygous mutants, namely *map3kε1*/−; *map3kε2*/+ and *map3kε1*/+; *map3kε2*/− were selected for further phenotypic analyses. Their vegetative growth showed no significant difference, compared to that of wild-type plants (Appendix A). However, the silique lengths of the *map3kε1*/−; *map3kε2*/+ and *map3kε1*/+; *map3kε2*/− mutants were significantly reduced, with an average of 8.0 mm (*n* = 107) and 9.6 mm (*n* = 162), respectively, much shorter than 12.00 mm (*n* = 100) in wild-type plants (Appendix A). The mutants produced an average of 34.0 and 37.6 seeds per silique, respectively, in contrast to an average of 48.2 seeds per silique in the wild-type plants. (Appendix A). To investigate the causes for the reduced seed set, the mature pollen grains at the late tricellular stage were examined by scanning electronic microscope (SEM). The results showed that 50.1% (*n* = 2226) of the *map3kε1*/−; *map3kε2*/+ and 51.2% (*n* = 2497) of *map3kε1*/+; *map3kε2*/− mutant pollen grains exhibited an abnormal morphology (Figure 1A–F,V). Alexander’s staining assays showed that many pollen grains from the two heterozygous mutants had an abnormal staining pattern, compared to those in red from wild-type plants (Figure 1G–I). DAPI staining showed that many mature pollen grains from the two heterozygous mutants were dead, as demonstrated by their loss of DAPI staining (Figure 1J–O). The aniline blue staining showed that 26.8% (*n* = 2940) of the *map3kε1*/−; *map3kε2*/+ and 23.5% (*n* = 3029) of the *map3kε1*/+; *map3kε2*/− mutant pollen grains at the tricellular stage exhibited abnormal callose accumulation, compared to those in wild-type plants (Figure 1P–U,W,X).

Pollen germination in the *map3kε1*/−; *map3kε2*/+ and *map3kε1*/+; *map3kε2*/− mutants were examined, respectively. The mature pollen grains were collected from the two heterozygous mutants and wild-type plants and cultured on the pollen germination medium, respectively. The time-course of germination rate in vitro was made by statistic method. At 1 h after being cultured (HAC), 6% (*n* = 2620) of wild-type pollen grains germinated. In contrast, 17% (*n* = 1111) of the pollen grains from *map3kε1*/−; *map3kε2*/+ and 16% (*n* = 1098) of the pollen grains from the *map3kε1*/+; *map3kε2*/− germinated in vitro. At two HAC, 21% (*n* = 667) of wild-type pollen grains germinated, whereas germination rate of the pollen grains from *map3kε1*/−; *map3kε2*/+ and *map3kε1*/+; *map3kε2*/− mutants increased by 53% (*n* = 1209) and 44% (*n* = 1187), respectively. At six HAC, the germination rates of the pollen grains from the two double mutants were close to 70%, with no significant difference between wild-type plants and the two double mutants (Appendix A). These results indicated that the MAP3Kεs negatively regulated pollen germination.

### 2.3. The map3kε1 map3kε2 Double Mutant Pollen Grains Precociously Germinated in the Dehiscent Anthers

The microscope observation showed that the pollen grains on the stigma in the *map3kε1*/−; *map3kε2*/+ and *map3kε1*/+; *map3kε2*/− plants were less than those on the stigma in wild-type plants (Figure 2A–C). Interestingly in the *map3kε1*/−; *map3kε2*/+ and *map3kε1*/+; *map3kε2*/− mutants, the mature pollen grains at the late tricellular stage germinated precociously and produced pollen tube in the dehiscent anthers. In particular, 37.2% (*n* = 223) of anthers in *map3kε1*/−; *map3kε2*/+ and 35.3% (*n* = 260) of anthers in *map3kε1*/+; *map3kε2*/− exhibited precocious pollen germination, respectively, contrast to that no or rare precocious pollen germination was observed in the wild-type plants (Figure 2D–N). These results further indicated that the MAP3Kεs were negative regulator of pollen germination before the pollen grains were pollinated on the stigma.

### 2.4. MAP3Kε1/2 Interacted with MOB1A/1B

Our previous study found that *ndr2/4/5* and *mob1a/+; mob1b/−* mutants also had precocious pollen germination in dehiscent anthers [38]. To investigate the function of MAP3Kε1/2 proteins, according to the components in the Hippo pathway that existed in yeast and the phylogenetic analysis, MAP3Kε1/2 may be analogies of GCK/Ste20 in yeast and interacted with the conserved NDR proteins. To confirm this interaction, both yeast two-hybrid (Y2H) and luciferase complementation image (LCI)assays were performed, respectively. In the Y2H experiment, NDR2/4/5 could not interact with MAP3Kε1/2, respectively. LCI assays showed that NDR2 and NDR4 could interact with MAP3Kε1/2, respectively, whereas NDR5 could not interact with MAP3Kε1/2 (Appendix A). These results implied that MAP3Kε1/2 maybe not interoperate with NDR2/4/5.

Previous studies showed that CGK/Ste20 protein can interact with the MOB-NDR complex [31,32], we investigated whether MAP3Kε1/2 could interact with MOB1A/1B. Three methods for protein interaction assays, namely Y2H, co-immunoprecipitation (Co-IP), and LCI assays, were applied. The Y2H results showed that MAP3Kε1/2 proteins could interact with MOB1A and MOB1B proteins in yeast cells, respectively (Figure 3A). Co-IP results showed that MOB1A-mcherry and MOB1B-mcherry could be pulled down by GFP-MAP3Kε1 and GFP-MAP3Kε2, respectively, indicating that MAP3Kε1/2 proteins could interact with MOB1A/1B proteins in vivo (Figure 3B). Finally, LCI analysis also showed that MAP3Kε1/2 could interact with MOB1A and MOB1B in *N. benthamiana* leaves, respectively (Figure 3C–J). The results demonstrated that MAP3Kε1/2 could interact with MOB1A and MOB1B.

### 2.5. MAP3Kε1/2 Were Co-Localized with MOB1A/1B in the Cytoplasm and Plasma Membrane

To investigate whether the MAP3Kεs proteins could be co-localized with the MOB1s proteins, the two p*UBQ10*::*eGFP*-*MAP3Kε* constructs were co-transformed into wild-type plants with p*UBQ10*::*MOB1A*-*mcherry* or p*UBQ10*::*MOB1B*-*mcherry*, respectively. The results showed that eGFP-MAP3Kε1 signals were co-localized in the cytoplasm, and plasma membrane with the signals of MOB1A-mcherry and MOB1B-mcherry, respectively. Similar results were also achieved from the assays for MAP3Kε2, showing that MAP3Kε2 was co-localized with MOB1A/1B in the cytoplasm, and plasma membrane (Figure 4A–H). These results also suggested that MAP3Kε1/2 interacted with MOB1A/1B and probably function in the cytoplasm and plasma membrane.

### 2.6. JA Enhanced Pollen Germination

A recent study revealed that *AtMOB1s* regulated JA accumulation subject to the *MYC2*-mediated JA signaling [37]. Therefore, we further investigated the expression levels of the genes in JA synthesis or signal pathway in the *map3kε1*/−; *map3kε2*/+ and *map3kε1*/+; *map3kε2*/− mutants. For instance, *LOX3*, *LOX4*, *LOX6*, *AOS*, *AOC1*, *AOC2*, *OPR3*, *CYP94B3*, *JOX3*, and *JAR1* which are involved in JA biosynthesis and metabolism, were significantly upregulated in the *map3kε1/−; map3kε2/+* and *map3kε1/+; map3kε2/−* mutants (Figure 5A). *JAZ1*, *JAZ2*, *JAZ5*, *JAZ9*, *JAZ10*, and *MYC2* in the JA signal pathway were also upregulated in the *map3kε1*/−; *map3kε2*/+ and *map3kε1*/+; *map3kε2*/− mutants (Figure 5B). Then, endogenous JA contents were measured in the mutant plants using the ultra-performance liquid chromatography (UPLC) combined with mass spectrometry. The JA content in the *map3kε1*/−; *map3kε2*/+ or *map3kε1*/+; *map3kε2*/− double mutant was increased compared with that in wild-type plants (Figure 5C). These results indicated that MAP3Kε1 and MAP3Kε2 maybe negatively regulate JA accumulation to maintain the homeostasis of JA level for restriction precocious pollen germination.

### 2.7. Genetic Interaction between MAP3Kε1/2 and MOB1A/1B

To explore the genetic interaction of *MAP3Kε1*/*2* and *MOB1A*/*1B*, we generated two types of multiple mutants: the *map3kε1*/−; *map3kε2*/+; *mob1a*/+; *mob1b*/− and *map3kε1*/+*; map3kε2*/−; *mob1a*/+; *mob1b*/− mutants, using the *map3kε1*/−; *map3kε2*/+ or *map3kε1*/+; *map3kε2*/− mutant plants as female to cross with *mob1a*/+; *mob1b*/− plant, respectively. The phenotypic analysis showed that these mutants also exhibited precocious pollen germination in dehiscent anthers. In particular, 80% (*n* = 157) dehiscent anthers of the *map3kε1*/−; *map3kε2*/+; *mob1a*/+; *mob1b*/− mutant and 93.5% (*n* = 184) dehiscent anthers of the *map3kε1*/+; *map3kε2*/−; *mob1a*/+; *mob1b*/− mutant showed precocious pollen germination, similar to 87% (*n* = 216) anthers with precocious pollen germination in the *mob1a*/+; *mob1b*/− plant (Figure 6A–M). Meanwhile, we also measured JA contents in the *map3kε1*/−; *map3kε2*/+; *mob1a*/+; *mob1b*/− and *map3kε1*/+; *map3kε2*/−; *mob1a*/+; *mob1b*/−, respectively. JA contents were significantly higher than those in the *map3kε1*/−; *map3kε2*/+ and *map3kε1*/+; *map3kε2*/−, but not obviously different from that in the *mob1a*/+; *mob1b*/− (Figure 6N). These results indicated that *MAP3Kε1/2* work as the upstream factors to interact with *MOB1A*/*1B* in regulation of pollen germination.

### 2.8. Genetic Interaction between MYC2 and MOB1A/1B

Due to the JA function in different plant development and *MYC2* was a key regulator in the JA pathway. We analyzed the genetic interaction between the *myc2-2* and *mob1a/+*; *mob1b/−* mutants, and used the *mob1a/+*; *mob1b/−* mutant plant as a female to cross with the *myc2-2* plant. We performed characterization of the *myc2-2/−*; *mob1a/+*; *mob1b/−* triple mutants and the phenotypic showed 5.72% (*n* = 62) of pollen germination in anthers of the *myc2-2/−*; *mob1a/+*; *mob1b/−* mutants and 10.72% (*n* = 84) pollen germination in anthers of the *mob1a/+*; *mob1b/−* mutants (Figure 7). The precocious pollen germination of the *mob1a/+*; *mob1b/−* was slightly reduced by the *myc2* mutant in *myc2-2/−*; *mob1a/+*; *mob1b/−* mutants. These results suggested that genetic interaction between *MYC2* and *MOB1A/1B* regulated pollen germination probably by JA signaling.

## 3. Discussion

### 3.1. MAP3Kε1 and MAP3Kε2 Functioned Redundantly and Involved in Control of Pollen Germination before Pollen Grains Landed on the Stigmas

MAPK cascades consist of three types of kinases, namely MAP kinase kinase kinases (MAPKKKs or MAP3Ks), MAP kinase kinases (MAPKKs or MAP2Ks) and MAP kinases (MAPKs). The Arabidopsis genome encodes 80 MAPKKKs, 10 MAPKKs and 20 MAPKs [40]. In plants, MAPK cascades regulate growth, development, stress responses, and immunity [41]. During plant reproduction, MAPK cascades regulate floral architecture, anther development, ovule development, and pollen tube guidance [42,43,44,45,46]. Several members of the MAPK cascades have been functionally characterized in Arabidopsis. The YODA(YDA), also known as MAPKKK4, affects zygote elongation and apical–basal polarity of the embryo [47] and the YDA-MKK4/5-MPK3/6 cascade is involved in regulating stomatal development, zygote elongation, asymmetric division of the zygote, and early embryo proper [41]. Arabidopsis NPK1-RELATED PROTEIN KINASE 1/2/3 (ANP1/2/3) belong to MAPKKKs, and the *anp2 anp3 mapk4* triple-mutant combination is lethal to female gametophyte [48]. ANP3-MKK6-MPK4 cascade plays an important role in male-specific meiosis [49]. Nevertheless, very little is known about the roles of MAPKs in pollen development and germination. The molecular mechanism of MAPK involved pollen development and pollen tube growth remains largely unclear. MAP3Kεs belong to MAPK kinase kinase kinase A4 sungroup and are required for pollen viability [39,50]. In this study, we further demonstrated that loss of the *MAP3Kεs* function caused abnormal accumulation of callose, the elevation of JA, and precocious pollen germination in the dehiscent anthers, indicating that *MAP3Kεs* had a complex mechanism underlying sexual plant reproduction.

The MAP3Kε1 and MAP3Kε2 shared homology with the Cdc7 and Cdc15 in yeast, which were the key components in the Hippo signaling pathway [28]. Our previous study demonstrated that the AGC kinases NDR2/4/5 interacted with the MOB1A/1B proteins in Arabidopsis, respectively, suggesting that the Hippo signaling components might be also existed in plants and played role in sexual plant reproduction [38]. This study showed that MAP3Kε1/2 could interact with the MOB1A/1B proteins, but is unlikely to directly interact with NDR2/4/5, indicating that MAP3Kε1/2 is also possibly related to the Hippo signaling pathway and involved in pollen germination in Arabidopsis.

### 3.2. MAP3Kε1/2 Controlled Pollen Germination Possibly by Crosstalk with JA Signaling

The biosynthesis of JA and the expression of JA-dependent genes can be regulated by MAPKs. For example, cosilencing of the *LeMPK1* and *LeMPK2* reduces JA biosynthesis and the expression levels of JA-dependent defense genes in tomato [51]. Overexpression of *OsMPK1* in rice leads to the accumulation of JA for disease resistance [52]. *AtMPK9* and *AtMPK12* in Arabidopsis are jointly involved in JA-induced stomatal closure [53,54]. *OsMPK12* and *OsMEK2* are activated by JA treatment during the defense response [55]. Besides, studies also show that JA is required for anther and pollen development [22,56]. In this work, we found that MAP3Kεs may regulate pollen germination through crosstalk with JA. We found the JA content was greatly increased in the *map3kε1*/−; *map3ε2*/+ and *map3kε1*/+; *map3ε2*/− mutants (Figure 5C). Moreover, the expression of essential genes in JA biosynthetic, metabolic, and signaling pathway, such as *LOX3*, *AOC2*, *JOX3*, and *JAZ*s, were apparently upregulated in the *map3kε1*/−; *map3ε2*/+ and *map3kε1*/+; *map3ε2*/− mutants (Figure 5A,B). In vitro pollen germination assays showed that application of 100 μM MeJA in the culture media could obviously enhance the germination rates of pollen, compared to the control medium (Appendix A). Recent study showed that disruption of *MYC2* partially rescued the root developmental defect and JA hypersensitivity of *mob1a/b* mutant [37]. As well as, *MYC2*, *MYC3*, *MYC4*, and *MYC5* were functionally redundant in stamen development and seed production [57]. We also performed characterization of the *myc2-2*/−; *mob1a*/+; *mob1b*/− triple mutant, and found that precocious pollen germination of the *mob1a*/+; *mob1b*/− mutant could be slightly recovered by the *myc2-2* mutation in *myc2-2*/−; *mob1a*/+; *mob1b*/− (Figure 7), indicating JA played a crucial role in pollen germination. Therefore, our results suggested that *MAP3Kε1*/*2* function in control of pollen germination, possibly through crosstalk with JA signaling pathway. Nevertheless, studies were required to further illustrate the mechanisms underlying the interaction of *MAP3Kε1*/*2* with JA signaling in control of pollen germination.

### 3.3. Callose May Be Involved in Regulation of Pollen Germination

Many studies show that callose metabolism is very important for pollen germination and pollen tube growth. Callose is a β-1,3-linked homopolymer of glucan and plays an important role in a variety of processes in angiosperms, including plant development and stress response [58]. During microsporogenesis, the transient callose walls forms and wraps the pollen mother cells (PMCs) and their meiotic progeny (microspores), forming a tetrad. Then callose is hydrolyzed by β-1,3-glucanase secreted from the tapetum, releasing the free microspores [10]. In addition, the callose deposition also occurs during the pollen germination and pollen tube growth. Defects in the callose deposition could severely affect the microspore development and pollen tube growth [10,59]. For example, *agp6 agp11* and *fla14* mutants display unusual callose deposition in the pollen grains and have precocious pollen germination in anthers [11,12]. Arabidopsis *rtg* mutations also cause abnormal callose accumulation in the pollen grains, leading to precocious pollen germination in the anthers [8]. The mutations in the Arabidopsis *GSL1*/*GSL10*/*CalS9* and overexpression of *GSL2*/*CalS5* and *PLANTACYANIN* also result in abnormal callose accumulation in pollen grains and precocious pollen germination in the anthers [9,10,60]. Moreover, callose deposition can also be regulated by JA in plants. Arabidopsis OCP3 (Cationic peroxidase 3) is as one of JA negative regulators to decrease the callose deposition and weak defense for *Botrytis cinerea* and *Plectosphaerella cucumerina* [61]. During the defense of tomato against *Botrytis cinerea*, callose deposition requires a complete JA biosynthesis pathway [62,63]. Flg22 triggers a plant’s innate immune response process—a COI-dependent JA signaling pathway that inhibits callose deposition [64]. In this study, our results showed that callose also accumulated abnormally in the *map3kε1*/−; *map3ε2/*+ and *map3kε1*/+; *map3ε2*/− mutants (Figure 1P–U,W,X). The expression levels of callose synthetase genes, such as *GSL2*, *GSL5*, *GSL7*, *GSL9*, were significantly upregulated in the two double mutants (Appendix A). Moreover, the expression levels of 1,3-glucanase genes, which were detected highly in pollen, were appreciably downregulated in the *map3kε1*/−; *map3ε2*/+ and *map3kε1*/+; *map3ε2*/− mutants (Appendix A). These results implied that the precocious pollen germination in the dehiscent anthers of *map3kε1*/−; *map3ε2*/+ and *map3kε1*/+; *map3ε2*/− mutants were likely associated with the abnormal callose accumulation in the pollen grains.

In this work, we found the MAP3Kε1 and MAP3Kε2 may be redundantly involved in the restriction of pollen germination before pollen grains landed on stigmas. Furthermore, MAP3Kε1/2 could interact with MOB1A/1B proteins, suggesting that MAP3Kε1/2 may interact with MOB1A/1B and played important roles in pollen germination possibly through crosstalk with JA and callose. Therefore, these findings provided the implications of MAP3Kε1s in plant reproductive development and as potential materials in crop production.

## 4. Materials and Methods

### 4.1. Plant Materials and Growth Conditions

The wild-type and mutant plants of Arabidopsis thaliana used in this study were Colombian ecotypes (Col-0) background. The Arabidopsis T-DNA insertion mutants *map3kε1* (SALK_133360C), *map3kε2* (SALK_084747C), *mob1a-1* (GK-719G04), *mob1b-1* (SALK_062070C) were obtained from Arabidopsis Biological Resource Center (ABRC). All seeds were surface-sterilized with 8% bleach for 10 min, then washed five times with sterile water, and plated on Murashige and Skoog (MS) media with or without 25 mg/L hygromycin [65]. After 2 days at 4 °C, the plates were transferred to the growth chamber with 16 h light/8 h dark cycles at 22 °C for 7 days. Then seedlings were transferred into soil and grown under the same condition. Sequences of all the primers used in the assays are listed in Appendix A.

### 4.2. RT-PCR and qRT-PCR Analysis

The RNA samples from different tissues of Arabidopsis and tobacco leaves were extracted using a total RNA extraction kit (TIANGEN, DP441, Beijing, China). In total, 1 μg RNA was reversely transcribed into cDNA using the RNA reversal kit (Gene-star, A224, Beijing, China). The RT-PCR assays were performed using the Taq PCR Star Mix (Gene-star, A012, China) and the program: 95 °C for 3 min, with 35 cycles of 95 °C for 30 s, and 52 °C for 30 s, 72 °C for 1 min. *TUBULIN8* was used as internal control [66]. Real-time PCR assays were performed using ABI Step One system (Thermo Fisher, Waltham, MA, USA) with Power SYBR Green PCR master mix (Gene-star, A303, Beijing, China) and followed by instructions of supplier’s protocol, with the following program: 95 °C for 10 min, with 40 cycles of 95 °C for 15 s, and 60 °C for 1 min. All PCR reactions were performed in triplicates. *ACTIN2* as internal control [65,67]. Sequences of all the primers used in the assays are listed in Appendix A.

### 4.3. Molecular Cloning and Transformation

To analyze gene expression patterns, the promoter fragments of *MAP3Kε1* and *MAP3Kε2* genes were amplified by PCR using the gene-specific primer pairs (Appendix A). The resulting promoter fragments were subcloned upstream of the GUS reporter gene in the modified pCAMBIA1300 vector (CAMBIA, Canberra, Australia) and introduced into wild-type plants using the *Agrobacterium*-mediated infiltration method [68]. GUS staining analysis was performed as previously described [69]. The transformants were screened on MS medium containing 25 mg/L hygromycin and used for evaluation of the complementation by phenotypic and genetic characterization. Sequences of all the primers used in the assays are listed in Appendix A.

### 4.4. Phenotypic Analyses

Morphology of the pollen grains was observed using scanning electron microscopy (TM4000, HITACHI, Tokyo, Japan) [70]. The pollen viability was assayed by Alexander’s staining as described in previous work [71]. Then, 4′6-diamidino-2-phenylindole (DAPI) staining of pollen grains was performed to visualize the nuclei in pollen grains as described previously [72,73]. Aniline blue is a specific dye for callose. The pollen grains or anthers were stained with a solution containing 0.1% aniline blue in 0.1 M K_2_HPO_4_-KOH buffer (pH = 11.0) for 5–10 min. Fluorescent images were obtained with a Leica DM2500 microscope (Leica, Wetzlar, Germany) under UV light [38].

For pollen in vitro germination assay, pollen grains were collected and spread on the pollen germination medium containing 20 mM CaCl_2_, 20 mM Ca(NO_3_)_2_, 20 mM MgSO_4_, 0.2%H_3_BO_4_, 1.8% sucrose, and 0.1% agarose. After incubation in the darkness at 28 °C, pollen germination was examined as previously described [66,74].

### 4.5. Yeast Two-Hybrid Assays

Yeast two-hybrid analysis was performed using Gal4 system vector (Clontech, https://www.takarabio.com/assets/a/112933, accessed on 20 February 2022). The coding sequences (CDSs) of *MAP3Kε1* and *MAP3Kε2* were cloned into the vector pGADT7 (AD), and *MOB1A* and *MOB1B* cDNA fragments were cloned into pGBKT7 (BD), respectively. Then constructs were co-transformed into yeast strain AH109 and then grow on SD/−Leu-Trp plates at 28 °C for 3 days. The colonies were identified by PCR, and positive colonies then continued to culture until the OD_600_ was 0.4~0.6, and transferred to grow on SD/−Leu-Trp plates and SD/−Trp-Leu-His for 3–7 days at 28 °C, respectively [75]. Sequences of all the primers used in the assays are listed in Appendix A.

### 4.6. Firefly Luciferase Complementation Imaging Assay

The CDSs of *MOB1A* or *MOB1B* were fused to the upstream of N-Luc in the pCAMBIA-NLuc vector, and *MAP3Kε1* or *MAP3Kε2* was fused to the downstream of C-Luc in the pCAMBIA-CLuc vector, respectively. The resulting constructs were transformed into *Agrobacterium* strain GV3101, which cultured in the medium containing 10 mM MgCl_2_, 10 mM MES, pH 5.7, 200 μM Acetosyringone, and then infiltrated into *N. benthamiana* leaves. CLuc-GUS and GUS-Nluc were used as negative control, and STG1-Nluc and Cluc-RAR1 were used as positive control [76]. After infiltration for 2–4 days, 1 mM D-luciferin was sprayed on the *N. benthamiana* leaves and kept in the darkness for 5 min, and the cooling CCD imaging system was used to capture LUC images with 8 to 10 min exposure time [65,77].

### 4.7. Co-Immunoprecipitation Assay

The Co-IP assay in *N. benthamiana* leaves was performed as described [78]. The total proteins were extracted from *N. benthamiana* leaves expressing p*UBQ10*::*GFP*-*MAP3Kε1*/p*UBQ10*::*MOB1A*-*mcherry* or p*UBQ10*::*GFP*-*MAP3Kε1*/p*UBQ10*::*MOB1A*-*mcherry* constructs with IP buffer containing 50 mM HEPES, pH = 7.4, 50 mM NaCl, 0.1%Triton X-100, 10 mM EGTA, pH = 8.0, 1 mM NaF, 100 mM DTT and 1× cocktail. In total, 50 μL was used as input detection, and the rest was incubated gently with anti-GFP agarose beads (AlpaLife, Shenzhen, China) at 4 °C for 2 h. After incubation, the agarose beads were washed five times with IP buffer and boiled for 10 min. The immunoprecipitated samples were finally separated on a 10% (*w*/*v*) SDS-PAGE and subjected to immunoblotting analysis with anti-GFP (EASYBIO, Beijing, China) and anti-mcherry antibody (EASYBIO, Beijing, China), respectively. The resulting images were recorded using the Fusion FX6 (Vilber, Paris, France) system. Further, p*UBQ10*::*GFP*-*MAP3Kε1*/p*UBQ10*::*MOB1B*-*mcherry* or p*UBQ10*::*GFP*-*MAP3Kε1*/*2*/p*UBQ10*::*MOB1B*-*mcherry* combinations were as described above.

### 4.8. Subcellular Localization

For the location of proteins, the full-length cDNA sequences of *MAP3Kε1* and *MAP3Kε2* were fused to the pCAMBIA1390-GFP vector by homologous recombination (Transgen, CU201-02, Beijing, China), and the full-length CDSs of *MOB1A* and *MOB1B* were fused to the pCAMBIA1390-mcherry vector. The resulting constructs were introduced into wild-type (WT) plants. The transgenic plants from both GFP-MAP3Kε1 and MOB1A-mcherry were selected in MS media containing 25 mg/L hygromycin. We observed the root using confocal laser scanning microscopy (CLSM, Zeiss LSM880 META microscope, Carl Zeiss, Oberkochen, Germany, https://www.zeiss.com.cn/microscopy/products/confocal-microscopes.html, accessed on 20 February 2022) data and collected images. The GFP signals at 500–550 nm were collected using the excitation light at 488 nm. The mcherry signals were excited at 561 nm and collected at 590–665 nm [65]. Additionally, GFP-MAP3Kε1/MOB1B-mcherry or GFP-MAP3Kε1/2 and MOB1B-mcherry combinations were as described above.

### 4.9. Measurement of JA and Callose

The flowers at the floral developmental stage 13 from wild-type (WT), *map3kε1*/−; *map3kε2*/+, *map3kε1*/+; *map3kε2*/−, *mob1a*/+; *mob1b*/−, *map3kε1*/−; *map3kε2*/+; *mob1a*/+; *mob1b*/− and *map3kε1/*+; *map3kε2*/−; *mob1a*/+; *mob1b*/− plants were collected, respectively, and placed in liquid nitrogen for quick-freezing preservation. In total, 50 mg powder was transferred into a 2 mL centrifuge tube, extracted with isopropanol solution, and then chloroform. The organic phase was finally blown dry at room temperature with nitrogen, and then solubilized with 0.1 mL of methanol before detection. The determination of JA was performed by ultraperformance liquid chromatography (UPLC) coupled with mass spectrometry [79].

Measurement of pollen callose was performed as described by Jones et al. [80]. Fluorescent signals were collected using a Spark^®^ (Zurich, Switzerland, https://lifesciences.tecan.cn/multimode-plate-reader, accessed on 20 February 2022). The excitation wavelength was 400 nm, the emission wavelength was 500 nm.

## Figures and Tables

**Figure 1 ijms-23-02683-f001:**
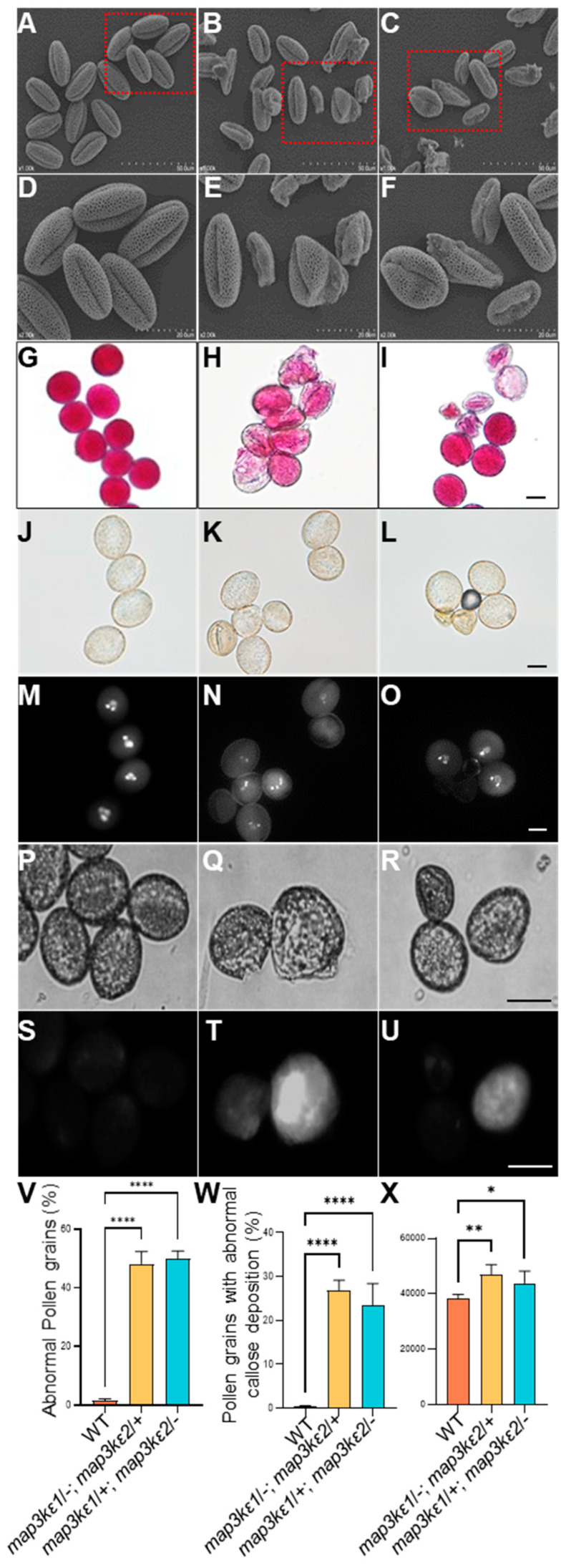
Characterization of the pollen grains in the *map3kε1*/−; *map3kε2*/+ and *map3kε1*/+; *map3kε2*/− mutants. (**A**–**C**): SEM images of mature pollen grains from the dehiscent anthers of wild-type (WT) (**A**), *map3kε1*/−; *map3kε2*/+ (**B**) and *map3kε1*/+; *map3kε2*/− (**C**) plants. (**D**–**F**): The magnified images of the red-boxed areas in (**A**–**C**), respectively. (**G**–**I**): The Alexander-stained pollen grains from wild-type (WT) plants (**G**), *map3kε1*/−; *map3kε2*/+ (**H**), and *map3kε1*/+; *map3kε2*/− (**I**) plants. (**J**–**O**): The DAPI stained pollen grains from wild-type (WT) (**J**,**M**), *map3kε1*/−;*map3kε2*/+ (**K**,**N**) and *map3kε1*/+; *map3kε2*/− (**L**,**O**) plants, where the bright field images (**J**,**K**,**L**) and fluorescent images (**M**–**O**) were shown, respectively. (**P**–**U**): Aniline blue-stained mature pollen grains from wild-type (WT) (**P**,**S**), *map3kε1*/−; *map3kε2*/+ (**Q**,**T**) and *map3kε1*/+; *map3kε2*/− (**R**,**U**) plants, where the bright field images (**P**–**R**) and fluorescent images (**S**–**U**) were shown, respectively. (**V**): Comparison of abnormal pollen grains in wild-type (WT), *map3kε1*/−; *map3kε2*/+ and *map3kε1*/+; *map3kε2*/−. (**W**): Comparison of pollen grains with callose deposition in wild-type (WT), *map3kε1*/−; *map3kε2*/+ and *map3kε1*/+; *map3kε2*/−. (**X**): Comparison of callose content in the pollen grains from wild-type (WT), *map3kε1*/−; *map3kε2*/+ and *map3kε1*/+; *map3kε2*/−, each bar represents the mean ± SD of three independent experiments. Different letters represent significant difference at * *p* < 0.05, ** *p* < 0.01, **** *p* < 0.0001 (one-way ANOVA, Tukey post-test). Bars = 10 μm in (**G**–**U**).

**Figure 2 ijms-23-02683-f002:**
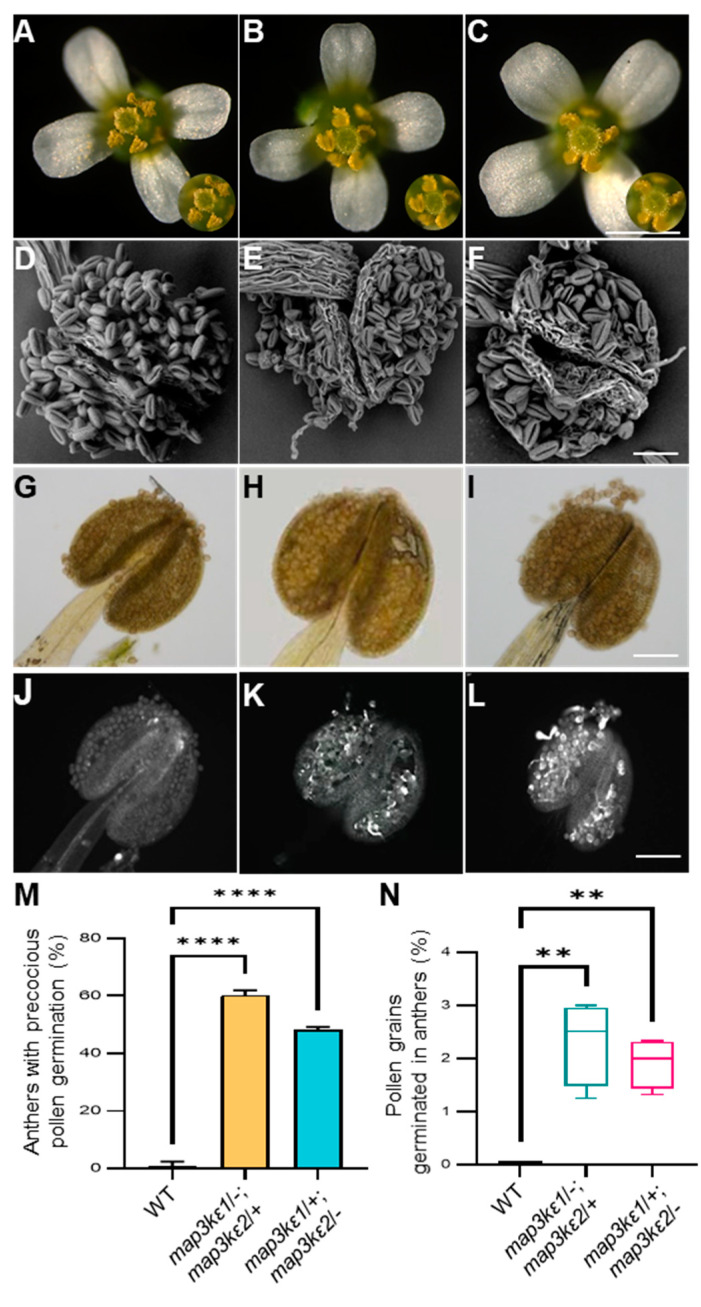
The *map3kε1*/−; *map3kε2*/+ and *map3kε1*/+; *map3kε2*/− exhibited precocious pollen germination in the dehiscent anthers. (**A**–**C**): The flowers at the floral stage 13 of wild-type (WT) (**A**), *map3kε1*/−; *map3kε2*/+ (**B**), and *map3kε1*/+; *map3kε2*/− (**C**) plants. (**D**–**F**): SEM images of the dehiscent anthers of wild-type (WT) (**D**), *map3kε1*/−; *map3kε2*/+ (**E**), and *map3kε1*/+; *map3kε2*/− (**F**) plants. (**G**–**L**): The aniline blue-stained dehiscent anthers of wild-type (WT) (**G**,**J**), *map3kε1*/−; *map3kε2*/+ (**H**,**K**), and *map3kε1*/+; *map3kε2*/− (**I**,**L**) plants, where the bright field images (**G**–**I**) and fluorescent images (**J**–**L**) were shown, respectively. (**M**): A comparison of the dehiscent anthers with precocious pollen germination in wild-type (WT), *map3kε1*/−; *map3kε2*/+ and *map3kε1*/+; *map3kε2*/−. (**N**): A comparison of precociously germinating pollen grains in the dehiscent anthers from wild-type (WT), *map3kε1*/−; *map3kε2*/+ and *map3kε1*/+; *map3kε2*/− plants. Different letters represent significant difference at ** *p* < 0.01, **** *p* < 0.0001 (one-way ANOVA, Tukey post-test). Bars = 1 mm in (**A**–**C**), 100 μm in (**D**–**L**).

**Figure 3 ijms-23-02683-f003:**
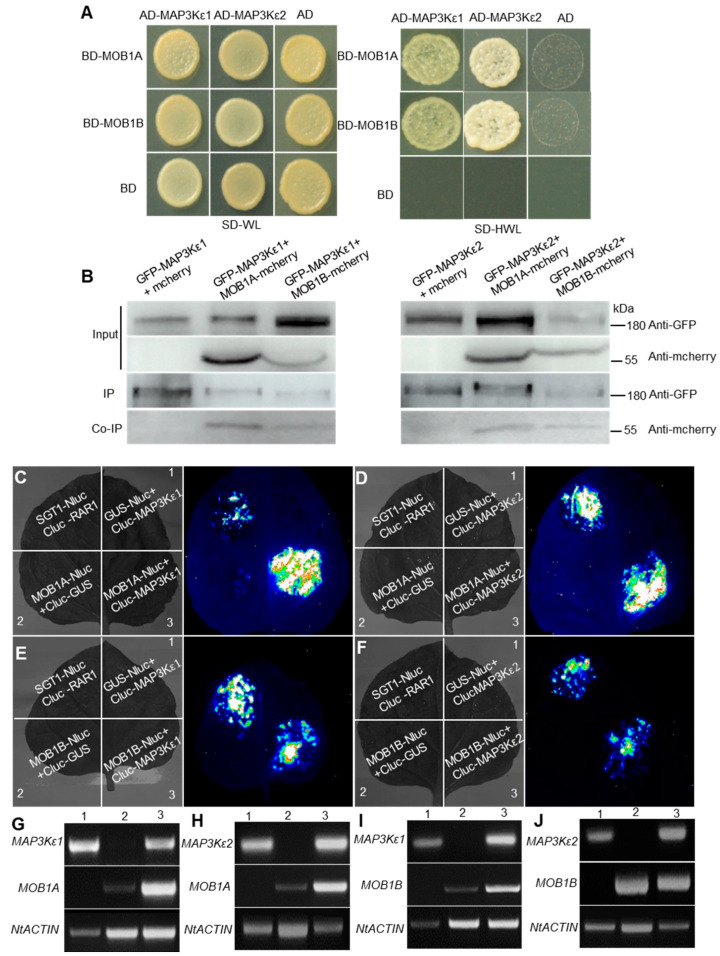
MAP3Kεs proteins interacted with the MOB1s proteins. (**A**): The MAP3Kεs proteins interacted with MOB1s proteins in the yeast two-hybrid assay. The interaction was determined by yeast growth on the medium lacking His (H), Trp (W), and Leu (L). (**B**): Co-IP of MAP3Kεs proteins and MOB1s proteins in vivo. The *N. benthamiana* leaves were transfected with p*UBQ10*::*GFP*-*MAP3Kε1*/*2* and p*UBQ10*::*mcherry* or p*UBQ10*::*GFP*-*MAP3Kε1*/*2* and p*UBQ10*::*MOB 1A*/*1B*-*mcherry*. The total protein extracts were immunoprecipitated with GFP Sepharose beads. Input and immunoprecipitated proteins were detected with anti-GFP and anti-mcherry antibodies. (**C**–**F**): MAP3Kεs proteins interacted with MOB1s proteins in the luciferase complementation image (LCI) assays. (**G**): RT-PCR analysis for gene expression in the LCI assay of (**C**). The numbers represent different experimental combinations marked in (**C**). (**H**): RT-PCR analysis for gene expression in the LCI assay of (**D**). The numbers represent different experimental combinations marked in (**D**). (**I**): RT-PCR analysis for gene expression in the LCI assay of (**E**). The numbers represent different experimental combinations marked in (**E**). (**J**): RT-PCR analysis for gene expression in the LCI assay of (**F**). The numbers represent different experimental combinations marked in (**F**). The expression of *N. benthamiana* gene *ACTIN* as control for the internal standard.

**Figure 4 ijms-23-02683-f004:**
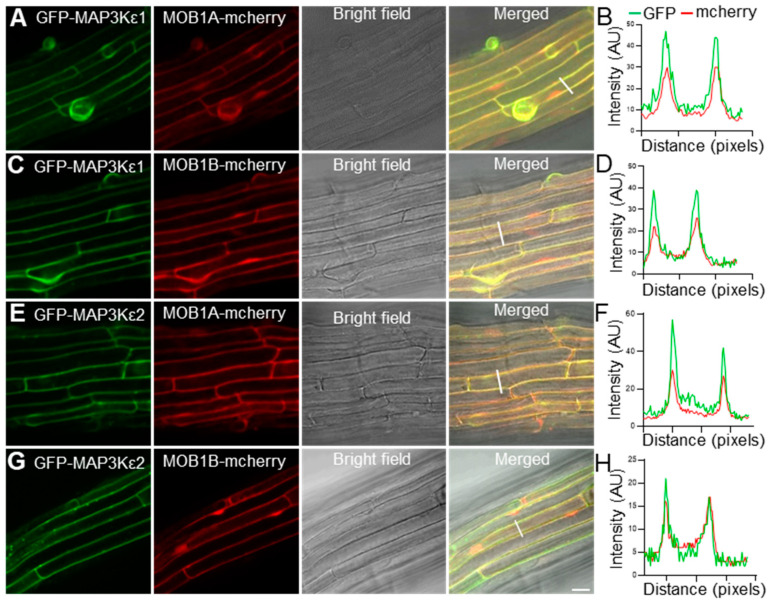
Subcellular localization patterns of MAP3Kεs and MOB1s proteins. (**A**): The fluorescent images of a root from the p*UBQ10*::*eGFP*-*MAP3Kε1* and p*UBQ10*::*MOB1A*-*mcherry* co-transgenic plant. (**B**): The fluorescence intensity distribution in the crossed regions (white lines) of (**A**). (**C**): The fluorescent images of a root from the p*UBQ10*::*eGFP*-*MAP3Kε1* and p*UBQ10*::*MOB1B*-*mcherry* co-transgenic plant. (**D**): The fluorescence intensity distribution in the crossed regions (white lines) of (**B**). (**E**): The fluorescent images of a root from the p*UBQ10*::*eGFP*-*MAP3Kε2* and p*UBQ10*::*MOB1A*-*mcherry* co-transgenic plant. (**F**): The fluorescence intensity distribution in the crossed regions (white lines) of (**E**). (**G**): The fluorescent images of a root from the p*UBQ10*::*eGFP*-*MAP3Kε2* and p*UBQ10*::*MOB1B*-*mcherry* co-transgenic plant. (**H**): The fluorescence intensity distribution in the crossed regions (white lines) of (**G**). Bars = 20 μm in (**A**,**C**,**E**,**G**).

**Figure 5 ijms-23-02683-f005:**
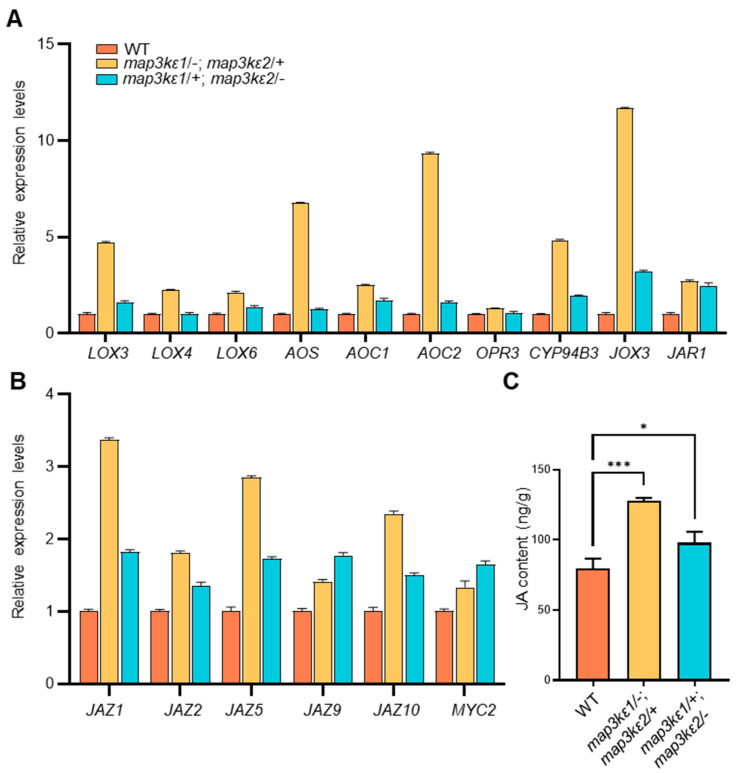
Expression of the genes in JA biosynthesis, metabolism, signaling, and JA contents were altered in the *map3kε1*/−; *map3kε2*/+, and *map3kε1*/+; *map3kε2*/− mutants. (**A**): Relative expression of JA biosynthesis, metabolism-related genes in mature pollen of wild-type (WT) and the *map3kε1*/−; *map3kε2*/+ and *map3kε1*/+; *map3kε2*/− mutant plants. (**B**): Relative expression of JA signaling-related genes in mature pollen of wild-type (WT) and the *map3kε1*/−; *map3kε2*/+ and *map3kε1*/+; *map3kε2*/− mutant plants. *ACTIN2* was used as the internal control. (**C**): Measurement of the JA contents in the opened flowers of wild-type (WT), *map3kε1*/−; *map3kε2*/+ and *map3kε1*/+; *map3kε2*/− mutants by using UPLC-MS. Each bar represents the mean ± SD of three independent experiments. Different letters represent significant difference at * *p* < 0.05, *** *p* < 0.001 (one-way ANOVA, Tukey post-test).

**Figure 6 ijms-23-02683-f006:**
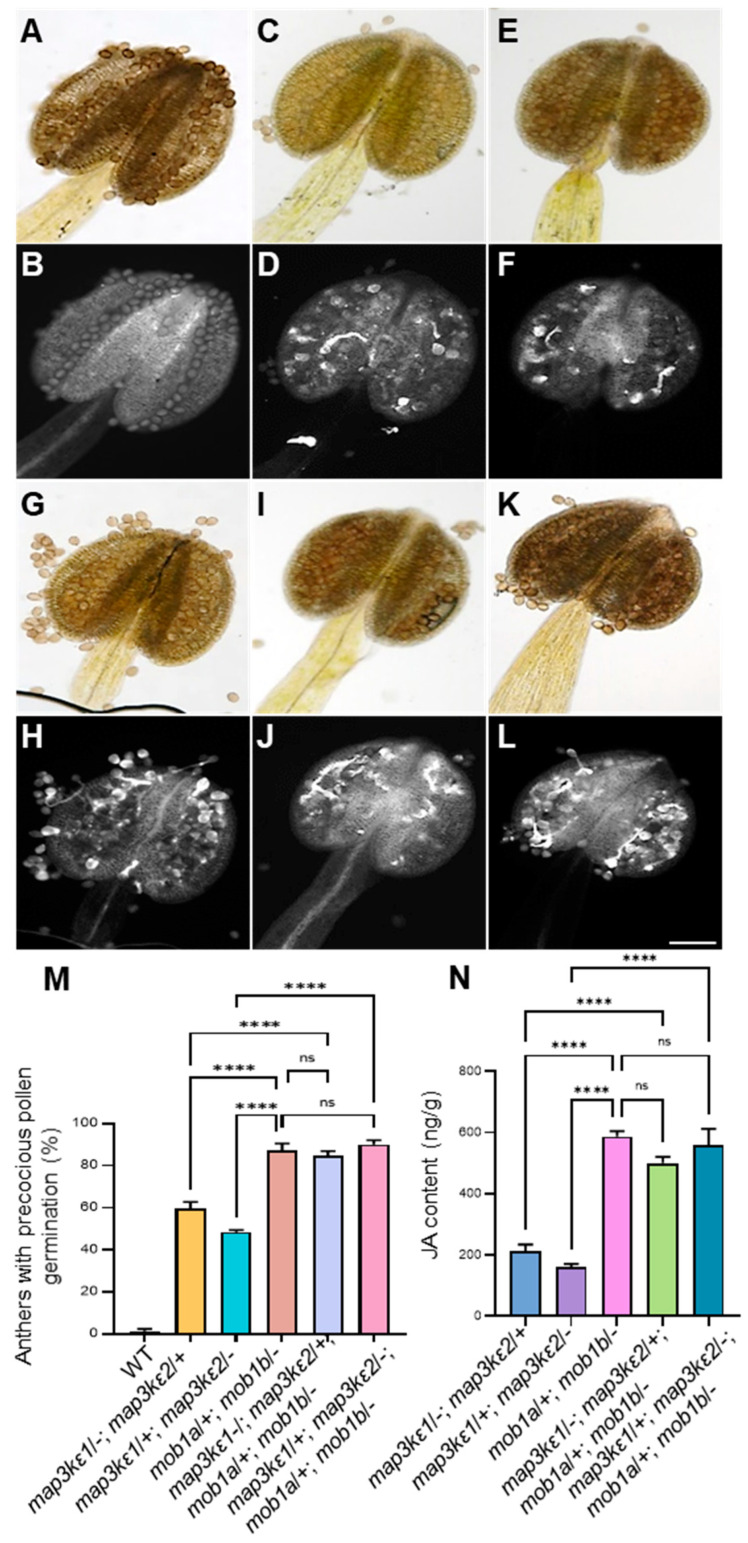
Genetic interaction of *MAP3Kε1*/*2* and *MOB1A*/*1B*. (**A**,**B**): The aniline blue-stained dehiscent anthers of wild-type (WT) plants, showing the bright field image (**A**), the fluorescent image (**B**). (**C**,**D**): The aniline blue-stained dehiscent anthers of *map3kε1*/−; *map3kε2*/+ plants, showing the bright field image (**C**), the fluorescent image (**D**). (**E**,**F**): The aniline blue-stained dehiscent anthers of *map3kε1*/+; *map3kε2*/− plants, showing the bright field image (**E**), the fluorescent image (**F**). (**G**,**H**): The aniline blue-stained dehiscent anthers *mob1a*/+; *mob1b*/− plants, showing the bright field image (**G**), the fluorescent images (**H**). (**I**,**J**): The aniline blue-stained dehiscent anthers of *map3kε1*/−; *map3kε2*/+; *mob1a*/+; *mob1b*/− plants, showing the bright field image (**I**), the fluorescent image (**J**). (**K**,**L**): The aniline blue-stained dehiscent anthers of *map3kε1*/+; *map3kε2*/−; *mob1a*/+; *mob1b*/−plants, showing the bright field image (**K**), the fluorescent image (**L**). (**M**): Comparison of anthers with precocious pollen germination of wild-type (WT), *map3kε1*/−; *map3kε2*/+, *map3kε1*/+; *map3kε2*/−, *mob1a*/+; *mob1b*/−, *map3kε1*/−; *map3kε2*/+; *mob1a*/+; *mob1b*/−, *map3kε1*/+; *map3kε2/−; mob1a*/+; *mob1b*/−. (**N**): Measurement of the JA contents in the opened flowers of *map3kε1*/−; *map3kε2*/+, *map3kε1*/+; *map3kε2*/−, *mob1a*/+; *mob1b*/−, *map3kε1*/−; *map3kε2*/+; *mob1a*/+; *mob1b*/− and *map3kε1*/+; *map3kε2*/−; *mob1a*/+; *mob1b*/− mutants by using UPLC-MS, each bar represents the mean ± SD of three independent experiments. Different letters represent significant difference at ns = not significant, **** *p* < 0.001 (one-way ANOVA, Tukey post-test). Bars = 100 μm in (**A**–**L**).

**Figure 7 ijms-23-02683-f007:**
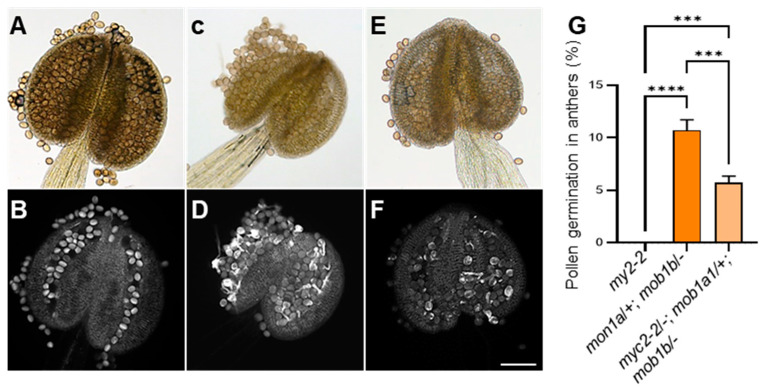
Genetic interaction of *MYC2* and *MOB1A/1B*. (**A**,**B**): The aniline blue-stained dehiscent anthers of *myc2-2* plants, showing the bright field image (**A**), the fluorescent image (**B**). (**C**,**D**): The aniline blue-stained dehiscent anthers of *mob1a*/+; *mob1b*/− plants, showing the bright field image (**C**), the fluorescent image (**D**). (**E**,**F**): The aniline blue-stained dehiscent anthers of *myc2-2*/−; *mob1a*/+; *mob1b*/− plants, showing the bright field image (**E**), the fluorescent image (**F**). (**G**): A comparison of precocious pollen germination in the dehiscent anthers from *myc2-2*, *mob1a*/+; *mob1b*/−, *myc2-2*/−; *mob1a*/+; *mob1b*/−, each bar represents the mean ± SD of three independent experiments. Different letters represent significant differences at *** *p* < 0.001, **** *p* < 0.0001 (one-way ANOVA, Tukey post-test). Bars = 100 μm in (**A**–**F**).

## Data Availability

We do not report additional data.

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
