# Peer review of "MAP3Kε1/2 Interact with MOB1A/1B and Play Important Roles in Control of Pollen Germination through Crosstalk with JA Signaling in Arabidopsis"

_ijms, 2022, doi:10.3390/ijms23052683_

Round 1

Reviewer 1 Report

The mechanisms, controlling the pollen germination in Arabidopsis were explained by functional characterization of the MAPKKK kinases (MAP3Kε1 and MAP3Kε2) in the current study. The two genes were expressed in the different tissues with higher expression levels in the tricellular pollen grains. They also mutated in Arabidobsis and analyzed the callose and JA content in the mutant plants. Interaction between MAP3Kε1/2 and MOB1A/1B were also investigated in the Hippo singling pathway of yeast. The results of study represented that MAP3Kεs interact with MOB1s and play important roles in restriction of the precocious pollen germination, possibly through crosstalk with JA signaling in Arabidopsis.

The methodology of the article, the novelty of the results, and the presentation of the results were excellently prepared. The article has been prepared in fluent English and deserves to be published in the journal as it is stand

Author Response

Thank you very much for the comments.

Reviewer 2 Report

 Brief Summary:

The manuscript “MAP3Kε1/2 interact with MOB1A/1B and play important roles in control of pollen germination through crosstalk with JA signaling in Arabidopsis”  is a study showing the role of two mitogen active kinases (MAP3K1 and 2) together with MOB1A and B in controlling pollen germination in Arabidopsis. They showed that MAP3K and MOB1 interact and fine-tune pollen tube germination in the same pathway through controlling jasmonic acid levels.

Broad comments:

The paper is well written with some room for imrovements of the text specially in the introduction part. Regarding the findins of this study, although the role of MAP3K in pollen tube germination is not a novel discovery, however the genetic interaction between MAP3K1/2 and MOB1A/B in map3kε1/-; map3kε2/+;mob1a/+; mob1b/- and map3kε1/+; map3kε2/-; mob1a/+; mob1b/- mutants as well as their physical interaction is an intersting piece of information and can be of interest for the experts in the field. The experimental designs and number of replicates seem sound and properly performed.

Specific comments:

Line 45: “evidence” instead of “evidences”

Line 203: It would be better to mention the full names of the protein interaction assays e.g. luciferase complementation imaging (LCI).

Figure 4: Co-localization as was shown in this figure does not prove if MAP3K2 specifically interact with MOB1. The fact that they are both in cytosol is not enough to say that they physically co-localize.

Line 358: “very little has been known” to “very little is known”

Lines 393 to 398: The generation of myc2-2/-;mob1a/+;mob1b/-triple mutant was not reported in the results section. In my opinion and for the sake of text flow and structure, I would move this to the results section and introduce the experiment and the outcome there first and just discuss the results in the discussion section.

Author Response

1. Line 45: “evidence” instead of “evidences”

Response: Thanks. We have replaced the “evidences” in abstract with “evidence”.

2. Line 203: It would be better to mention the full names of the protein interaction assays e.g. luciferase complementation imaging (LCI).

Response: Thank you very much. We have written all abbreviation names of the protein interaction assays and others, such as the full name “luciferase complementation image assays” instead of “LCI” at the first appearance in Line 106.

3. Figure 4: Co-localization as was shown in this figure does not prove if MAP3K2 specifically interact with MOB1. The fact that they are both in cytosol is not enough to say that they physically co-localize.       

Response: Thank you for your suggestion. Due to the biochemical experiments in the part 2.4 of the “results section”, we have characterized that MAP3K2 interacted with MOB1, not specifically. Co-localization showed that they functioned probably similarly.

4. Line 358: “very little has been known” to “very little is known”   

Response: Thanks. We have corrected it as “very little is known” in the Line 358.

5. Lines 393 to 398: The generation of myc2-2/-;mob1a/+;mob1b/-triple mutant was not reported in the results section. In my opinion and for the sake of text flow and structure, I would move this to the results section and introduce the experiment and the outcome there first and just discuss the results in the discussion section.

Response: Thank you for your suggestion. We have moved “The generation of myc2-2/-;mob1a/+;mob1b/-triple mutant” as part 2.7 in the results section. We further explain the experiment and also discuss it slightly in the discussion section.

Reviewer 3 Report

In the manuscript, valuable results about factors influencing pollen germination are presented. The results are confirmed in many experiments which were well planned. I read this manuscript very carefully and in my opinion, it has any deficiencies. Nice job.

Author Response

Thank you very much for your comments.

Reviewer 4 Report

The Arabidopsis mob1a mob1b mutants also exhibited a similar phenotype of precocious pollen germination to that in map3kε1 map3kε2 mutants. MAP3Kε1 and MAP3Kε2 redundantly play important roles in control of pollen germination in Arabidopsis, possibly through crosstalk with JA signaling and influencing the callose accumulation in pollen grains. Although the overall interest and visibility of this work, some aspects should still be considered to improve the quality and objectiveness of this work. Overall, it is an important study and should be considered for publication once the issues have been resolved.

  1. Please speculate about the reasons for the obtained results. The discussion needs to improve.
  2. In Conclusion, the authors should add the potential practical application.
  3. The article should need to be reviewed for English language proficiency and grammar. There are a lot of sentences without sense, misspelled words, errors with punctuation.

Author Response

1. Please speculate about the reasons for the obtained results. The discussion needs to improve.

Response: Thanks for your suggestion. We review the relevant references and also add some information to improve the discussion in the manuscript.

2. In Conclusion, the authors should add the potential practical application.

 Response: Thanks. We have added the potential applications in the discussion and conclusion.

3. The article should need to be reviewed for English language proficiency and grammar. There are a lot of sentences without sense, misspelled words, errors with punctuation.

 Response: Thanks a lot. We revised the manuscript for grammar, words and punctuation.